# A comparative genomic study of a hydrocarbon-degrading marine bacterial consortium

**Jorge Rojas-Vargas**[1][¤], **Eria A. Rebollar**[2], **Alejandro Sanchez-Flores**[3], **Liliana Pardo-López**[1]*

**1** Departamento de Microbiología Molecular, Instituto de Biotecnología, Universidad Nacional Autónoma de México, Cuernavaca, Morelos, Mexico, **2** Programa de Microbiología Genómica, Centro de Ciencias Genómicas, Universidad Nacional Autónoma de México, Cuernavaca, Morelos, Mexico, **3** Instituto de Biotecnología, Unidad Universitaria de Secuenciación Masiva y Bioinformática, Universidad Nacional Autónoma de México, Cuernavaca, Mexico

¤ Current address: Department of Biology, University of Western Ontario, London, Ontario, Canada
* liliana.pardo@ibt.unam.mx

**Data Availability Statement:** The three new complete genomes published in this study have been deposited in GenBank under the following NCBI accession numbers: Alloalcanivorax xenomutans GOM8 (Accession No.

## Abstract

Ocean oil pollution has a large impact on the environment and the health of living organisms. Bioremediation cleaning strategies are promising eco-friendly alternatives for tackling this problem. Previously, we designed and reported a hydrocarbon (HC) degrading microbial consortium of four marine strains belonging to the species *Alloalcanivorax xenomutans*, *Halopseudomonas aestusnigri*, *Paenarthrobacter* sp., and *Pseudomonas aeruginosa*. However, the knowledge about the metabolic potential of this bacterial consortium for HC bioremediation is not yet well understood. Here, we analyzed the complete genomes of these marine bacterial strains accompanied by a phylogenetic reconstruction along with 138 bacterial strains. Synteny between complete genomes of the same species or genus, revealed high conservation among strains of the same species, covering over 91% of their genomic sequences. Functional predictions highlighted a high abundance of genes related to HC degradation, which may result in functional redundancy within the consortium; however, unique and complete gene clusters linked to aromatic degradation were found in the four genomes, suggesting substrate specialization. Pangenome gain and loss analysis of genes involved in HC degradation provided insights into the evolutionary history of these capabilities, shedding light on the acquisition and loss of relevant genes related to alkane and aromatic degradation. Our work, including comparative genomic analyses, identification of secondary metabolites, and prediction of HC-degrading genes, enhances our understanding of the functional diversity and ecological roles of these marine bacteria in crude oil-contaminated marine environments and contributes to the applied knowledge of bioremediation.

GCA_032810065.1), Paenarthrobacter sp. GOM3 (Accession No. GCA_018215265.2), Psedomonas aeruginosa GOM9 (Accession No. GCA_032811685.1). The Supporting Tables S1–S6 are available in https://figshare.com/articles/dataset/S1-S5_Tables/24571126, and the S1 Fig is available in https://figshare.com/articles/figure/S1_Fig/24571585.

**Funding:** JR-V was a doctoral student from the Programa de Doctorado en Ciencias Bioquímicas, Universidad Nacional Autónoma de México (UNAM), and received fellowship 965003 from the Consejo Nacional de Humanidades, Ciencia y Tecnología (CONACYT). This work was supported by UNAM-PAPIIT IG200223 and CONACYT – Mexican Ministry of Energy- Hydrocarbon Trust, project 201441. This is a contribution of the Consorcio de Investigación del Golfo de México (CIGoM).

**Competing interests:** The authors have declared that no competing interests exist.

## Introduction

Crude oil spills, a consequence of anthropogenic activities in the ocean, threaten marine ecosystems by contaminating waters and harming wildlife. Some of the oil's hydrocarbons (HCs) are recalcitrant, carcinogenic, or mutagenic. Nevertheless, they serve as a carbon and energy source for certain microorganisms, including marine bacteria. These HC-degrading bacteria play a crucial role in mitigating HC pollution in marine environments. Their biotechnological potential is increasingly being recognized, offering a natural and eco-friendly approach to bioremediation [1]. Research in this rapidly growing field has shed light on the biological processes that enable HC biodegradation [2,3].

HC-degrading marine bacteria have specialized genes for metabolizing these diverse compounds. Several of these genes and their encoded proteins have been identified and characterized through genomic and experimental studies. These include the alkane monooxygenase (*alkB*) and the flavin-binding protein monooxygenase (*almA*), involved in alkane degradation [4,5], and the catechol dioxygenase (*catA*) and the naphthalene 1,2-dioxygenase system (*ndoABCR*), involved in aromatic degradation [6–8]. The latter examples are just a fraction of the diverse genetic repertoire employed by these bacteria [9]. Several bacterial genera, including *Alcanivorax*, *Halomonas*, *Marinobacter*, and *Pseudomonas*, are frequently associated with marine HC degradation [10,11].

However, a single bacterium cannot fully metabolize the complex mixture of HCs found in crude oil. Effective degradation often relies on diverse bacterial communities working together [12,13]https://paperpile.com/c/Jxoarz/Vxfc+cKMm. Researchers have been exploring the use of natural or engineered consortia, groups of bacteria, to degrade oil more efficiently to be potentially used in bioremediation strategies such as bioaugmentation or the addition of microorganisms with oil-degrading capacity [14,15].

In this context, our research focuses on our previously laboratory-designed consortium of four marine bacteria isolated from the Gulf of Mexico (GoM) [15], a region naturally exposed to HCS. This consortium, specifically assembled to degrade light crude oil in seawater, includes bacteria of the *Alloalcanivorax* and *Pseudomonas* genera, along with *Paenarthrobacter* sp. GOM3 and *Halopseudomonas aestusnigri* GOM5. The first two were previously identified using 16S rRNA sequencing [15], and the latter two have been previously sequenced using NGS platforms [16,17].

Our consortium effectively removed 62% of light crude oil compared to a maximum of 39% by individual strains [15]. To gain a better understanding of their remarkable teamwork, we need to delve into their metabolic potential. By analyzing the complete genomes of each bacterium, we aim to identify genes encoding proteins crucial for HC metabolism and biosurfactant production and to compare their genomes with other bacteria within the same genera to understand how they evolved this genomic content. This research will provide valuable insights into the evolution and ecological role of these marine bacteria in oil-contaminated environments. Unraveling their unique genetic code and metabolic machinery will be a significant contribution to the field of bioremediation.

## Materials and methods

### Consortium strains and nucleic acid extraction

The four bacterial strains of the consortium were isolated from the GoM as previously described in [15]. Briefly, the isolation procedure involved culturing environmental samples on specific media to enrich for HC-degrading bacteria. Individual colonies were then purified

through repeated streaking on fresh media plates. Glycerol stocks of these bacteria were stored at -80˚C.

*Alloalcanivorax xenomutans* GOM8, *Paenarthrobacter* sp. GOM3 and *Pseudomonas aeruginosa* GOM9 were individually cultured from glycerol stocks overnight at 30˚C in 25 mL of LB medium in a 125 mL flask shaken at 180 rpm. Total DNA was extracted from each culture using a Quick-gDNA Miniprep Kit from Zymo Research (Irvine, CA, United States) following the kit instructions. Separate DNA extractions from different inocula were used for each sequencing technology. In the case of *Paenarthrobacter* sp. GOM3, genomic DNA was only extracted for use with Nanopore sequencing technology. At the time of this study, the complete genome of *H. aestusnigri* GOM5 was already available [17] and was therefore not included in the present DNA extraction step.

## Genome sequencing, assembly, and annotation

The complete genome of *H. aestusnigri* GOM5 (RefSeq accession No. GCF_021184005.1) and the Illumina raw reads of the *Paenarthrobacter* sp. GOM3 (SRA accession No. SRR26386755) were retrieved from the NCBI portal (https://www.ncbi.nlm.nih.gov/). To obtain the complete genomes of *Paenarthrobacter* sp. GOM3 and the other two bacteria, we applied the hybrid approach used for the GOM5 strain [17]. The completeness and contamination analysis was carried out with the CheckM software v1.1.3 using the taxonomy workflow with the genus option [18]. Gene predictions were conducted with RAST-tk [19], and functional annotations were performed using KofamKOALA [20] of the Kyoto Encyclopedia of Genes and Genomes (KEGG) [21]. Secondary metabolite-related gene clusters were predicted using antiSMASH v7.0 in strict mode [22]. Default parameters were used for all software unless otherwise specified.

## Identification of genes related to HC degradation

To identify genes associated with HC-degradation, we used Proteinortho [23] with the parameters -identity = 50 and -conn = 0.3 and compared ortholog groups against the HADEG database, as described in [9]. Subsequently, we complemented the HADEG gene prediction with the RAST-tk functional annotation to identify HC-degradation genes with an identity <50% against the sequences in HADEG. This strategy was used to obtain a more comprehensive description of the HC degradation genes present in genomes.

## Taxonomic assessment

For taxonomic assignment, we compared the four complete genomes with the collection of prokaryotic genomes included in the Type (Strain) Genome Server (TYGS) [24]. The top ten nonredundant genomes of type strains based on their digital DNA-DNA hybridization (dDDH) values were downloaded from the NCBI database (consulted on June 9, 2023). Additionally, TYGS was used to determine the percentage differences in G+C content among closely related strains. The average nucleotide identity values based on MUMmer (ANIm) were calculated using JSpeciesWS v3.9.0 web server [25] (http://jspecies.ribohost.com/jspeciesws/#analyse). To confirm the taxonomic classification of each bacterial strain within a specific species, they needed to meet the following criteria: ANI% >96%, dDDH >70%, and G+C content difference <1% [26].

## Phylogenetic analysis

For the phylogenomic reconstruction of the consortium strains, we obtained genomes from the genera *Alloalcanivorax*, *Halopseudomonas*, and *Paenarthrobacter* that met specific criteria:

from isolates, number of contigs <500, and completeness >90%, and contamination <1% according to the CheckM analysis in the GTDB database (https://gtdb.ecogenomic.org/). These genomes were retrieved from the NCBI database (accessed on June 9, 2023). We downloaded a total of 17 *Alloalcanivorax* genomes, 24 *Halopseudomonas* genomes, and 32 *Paenarthrobacter* genomes. For the *Pseudomonas* genus, we selected complete genomes with the same aforementioned features but limited our selection to type strains. Additionally, we included five more genomes from *P. aeruginosa* isolates sourced from marine environments for comparative purposes. A total of 60 *Pseudomonas* genomes were obtained from NCBI. As outgroup genomes, we used five genomes representing the phyla Firmicutes and Cyanobacteria: *Bacillus amyloliquefaciens* ATCC 23350 (GCF_000196735.1), *B. subtilis* subsp. subtilis str. 168 (GCF_000009045.1), *Allocoleopsis franciscana* PCC 7113 (GCF_000317515.1), *Gloeobacter violaceus* PCC 7421 (GCF_000011385.1), and *Microcystis aeruginosa* NIES-843 (GCF_000010625.1). The complete list of the 138 genomes used for phylogenomic reconstruction can be found in S1 Table.

A set of 92 bacterial core genes from the genomes of the consortium and the previously mentioned 138 genomes was extracted and aligned using UBCG v3.0 with a filtering cutoff of -f 90 for gap-containing positions [27]. The nucleotide substitution model was determined using IQ-TREE v2.0.3 using the -T AUTO parameter [28]. This software was also used to construct a maximum likelihood (ML) tree based on 1,000 ultrafast bootstrap replicates.

## Comparative genomics

For comparative genomic analysis, we selected four complete genomes of close relatives to the consortium bacteria. The genomes of *A. xenomutans* P40, *Halopseudomonas* sp. MFKK-1, *Paenarthrobacter nicotinovorans* ATCC 49919, and *P. aeruginosa* DSM 50071 were retrieved from the NCBI database (consulted on June 9, 2023). Table 1 provides a comprehensive list comparing their genomic features with our strains. To analyze gene order conservation or genome synteny between our bacteria and the selected strains, we used Sibelia software v3.0.7 using default parameters [29], and the resulting syntenic block alignments were visualized in Circos [30].

## Gene gain and loss analysis

We used the program COUNT [31] to estimate the number of gene gains and losses that likely occurred along the branches of the phylogenetic tree reconstructed with the 92 bacterial core

**Table 1. Genome features of the four consortium marine strains and the reference assemblies for comparative genomic analysis.** The consortium genome features are marked in gray.

| Organism | Genome features | | | | | | | | |
|---|---|---|---|---|---|---|---|---|---|
| | Genome Size (bp) | Contigs | CheckM completeness | CheckM contamination | G+C content (%) | CDS | tRNA | rRNA | Accession Number |
| *Alloalcanivorax xenomutans* GOM8 | 4,528,424 | 1 | 99.54 | 1.89 | 61.62 | 4,204 | 43 | 6 | GCA_032810065.1 |
| *A. xenomutans* P40 | 4,733,951 | 1 | 99.54 | 2.05 | 61.45 | 4,456 | 44 | 3 | GCA_002072815.1 |
| *Halopseudomonas aestusnigri* GOM5 | 3,996,286 | 1 | 94.26 | 1.88 | 60.61 | 3,747 | 58 | 12 | GCA_021184005.1 |
| *Halopseudomonas* sp. MFKK-1 | 3,839,705 | 1 | 93.43 | 2.00 | 60.82 | 3,521 | 59 | 12 | GCA_027474385.1 |
| *Paenarthrobacter* sp. GOM3 | 4,464,400 | 1 | 99.71 | 0.39 | 63.16 | 4,193 | 55 | 18 | GCA_018215265.2 |
| *Paenarthrobacter nicotinovorans* ATCC 49919 | 4,481,325 | 2 | 85.21 | 10.14 | 63.10 | 4,214 | 54 | 18 | GCA_021919345.1 |
| *Pseudomonas aeruginosa* GOM9 | 6,889,943 | 1 | 99.39 | 2.05 | 66.03 | 6,508 | 66 | 12 | GCA_032811685.1 |
| *P. aeruginosa* DSM 50071 | 6,317,050 | 1 | 99.37 | 0.49 | 66.52 | 5,946 | 65 | 4 | GCA_001045685.1 |

genes. As input for COUNT, we used the reconstructed ML tree and a table containing the number of orthologs identified within each genome. These orthologs were determined using Proteinortho with the same parameters employed for annotating HC-degradation genes and pathways. In COUNT, we applied the Wagner parsimony model with flexible gain-loss ratios for all lineages and used a Poisson distribution at the root.

## Results and discussion

### Genomic features of the marine consortium bacterial isolates

Genomes were *de novo* assembled, and circular genomes were obtained with high quality, completeness above 99%, and contamination below 1% (Table 1). The G+C content percentages were consistent with previously reported genomes for the respective species or genera of our strains, with between 60.61% and 66.03% G+C content. Among the consortium strains, *H. aestusnigri* GOM5 exhibited the smallest genome size at 3.99 Mbp, while *P. aeruginosa* GOM9 showed the largest genome size at 6.89 Mbp. Our updated version of the *Paenarthrobacter* sp. GOM3 genome incorporated an additional 45,401 bp to the first version (first version accession number GCA_018215265.1, genome size 4,418,999 bp), resulting in an increase of 70 genes, 5 CDSs, 3 tRNAs, and 12 rRNAs, according to the RAST-tk annotation.

### Taxonomic assignment of the marine consortium bacterial isolates

We calculated the average nucleotide identity based on MUMmer (ANIm), digital DNA-DNA hybridization (dDDH), and G+C content differences between the genomes of the four consortium bacteria and the closest type strain genomes (S2 Table). Applying prokaryotic species delineation criteria, strain GOM8 is closely related to *Alloalcanivorax xenomutans* JC109 (98.44% ANIm, 84.9% dDDH, and 0.12% G+C content difference) indicating its affiliation with this species. Similarly, strain GOM9 closely resembles *Pseudomonas aeruginosa* DSM 50071 (99.43% ANIm, 94.9% dDDH, and 0.45% G+C content difference), suggesting its classification within this species. Analysis of the GOM3 genome revealed the closest type strain genome to be *Paenarthrobacter nicotinovorans* DSM 420 (85.69% ANIm, 25.5% dDDH, and 0.05% G+C content difference), suggesting that it represents a novel species within the genus *Paenarthrobacter*, which confirms previous findings [16]. Strain GOM5 exhibits a close relationship with *Halopseudomonas aestusnigri* CECT 8317 (97.01% ANIm, 73.1% dDDH, and 0.31% G+C content difference), confirming its classification as *H. aestusnigri*.

### Genomic phylogenetic reconstruction

To explore the phylogenetic relationships among the bacteria within the marine consortium, we constructed a maximum likelihood tree using 92 marker genes found in the genomes of 138 prokaryotic bacteria (S2 Table), and our four strains (see "Materials and Methods" section). The tree revealed four distinct clades (Fig 1): one clade included the *Paenarthrobacter* genus, which belongs to the phylum Actinobacteria and formed a separate major branch, indicating a greater evolutionary distance from the other genera due to its distinct phylum affiliation. Notably, the genome of the marine strain *Paenarthrobacter* sp. GOM3 formed a subclade with two other genomes of novel *Paenarthrobacter* species, isolated from a US potato (strain UW852) and an Indian lake (strain AR 02), respectively. ANI comparisons indicated that UW852 and AR 02 share an ANI value of 96.70%, suggesting that they belong to the same species (S3 Table). However, when compared to the GOM3 assembly, ANI values range from 90.91% to 91.25%, indicating a low genetic similarity and suggesting that GOM3 represents a distinct new species within the *Paenarthrobacter* genus.

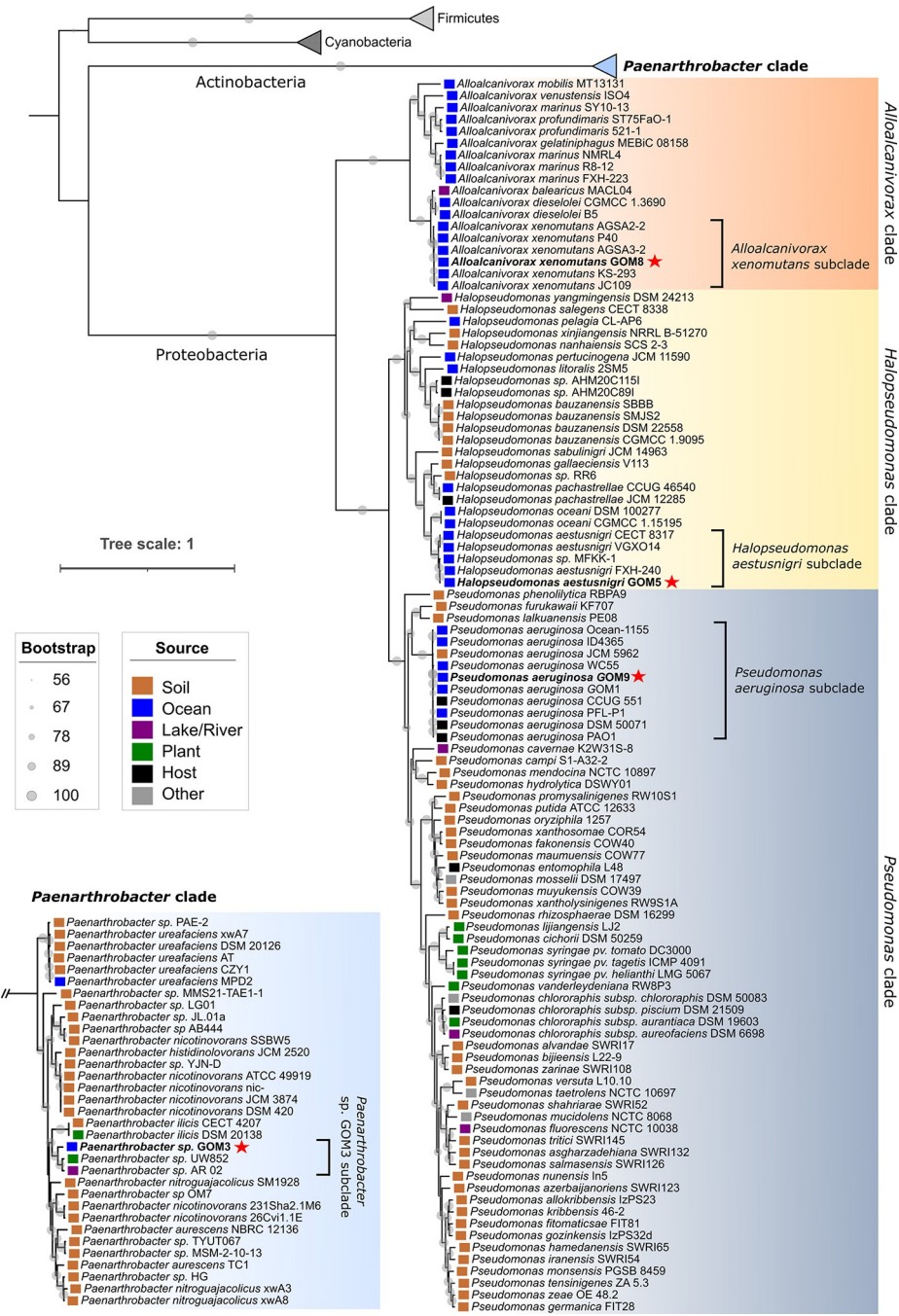

**Fig 1. Phylogenetic reconstruction using the 92 marker genes extracted with the UBCG tool for prokaryotes.** The tree was constructed using the maximum likelihood method with IQ-TREE and was based on 1,000 ultrafast bootstrap replicates. The obtained substitution model was TIM+F+R10. Clades corresponding to the four genera of the consortium genomes, namely *Alloalcanivorax* (orange), *Halopseudomonas* (yellow), *Paenarthrobacter* (pale blue), and *Pseudomonas* (dark blue), are highlighted. The consortium genomes are denoted in bold with a red star, and the origin of each strain is shown with colored squares.

The remaining three clades belonged to the genera *Alloalcanivorax*, *Halopseudomonas*, and *Pseudomonas*, all of which fall within the phylum Proteobacteria. The *A. xenomutans* GOM8 assembly clustered with five other marine genomes of the same species (*A. xenomutans*

subclade in Fig 1), all from seawater or sediment samples. Since these genes belong to the same species, these genomes shared ANI values above 96%, ranging from 98.30% to 99.35% (S3 Table). The genome of the *H. aestusnigri* GOM5 strain formed a subclade (*H. aestusnigri* subclade) with three other assemblies of the same marine species and *Halopseudomonas* sp. MFKK-1. The ANI values of the subclade ranged from 96.97% to 99.98%, suggesting a close genetic relatedness, confirming the inclusion of the MFKK-1 strain, isolated from Japanese seawater, within the *H. aestusnigri* species.

Finally, the *P. aeruginosa* GOM9 genome was clustered together with nine other assemblies of the same species (*P. aeruginosa* subclade), which were isolated from different environments, including hosts, soil, and the ocean. Comparative analysis of these assemblies revealed ANI values ranging from 98.74% to 99.99% (S3 Table). Notably, the GOM9 genome (6.9 Mbp) has the highest ANI value of 99.99% compared to the GOM1 genome (7.1 Mbp). This high genomic similarity between the orthologous genes of these strains likely stems from the fact that both strains were isolated from seawater samples collected in the GoM. Since these bacteria originated from the same environment, they likely experienced similar evolutionary pressures that shaped their genomes.

## Synteny comparison among complete genomes

We compared the genomic organization between the marine consortium genomes and other genomes of the same species or genus. Synteny conservation represents a higher relationship between taxa and potentially indicates the presence of physiologically important gene clusters. Disrupting synteny patterns could be caused by factors like horizontal gene transfer (HGT) or adaptation to specific ecological niches [32].

Our analysis revealed that the genomes of the same species shared a remarkable syntenic level, covering over 91% of their sequences (Fig 2). This trend was observed in the *A. xenomutans* GOM8 genome compared to the *A. xenomutans* P40 assembly, where 9 synteny blocks were aligned, covering over 98% of the genomes (Fig 2A). Similarly, *H. aestusnigri* GOM5 and *Halopseudomonas* sp. MFKK-1 genomes shared 15 out of 17 aligned blocks, with over 91% coverage (Fig 2B). As mentioned in the previous section, MFKK-1 could belong to the *H. aestusnigri* species according to the ANI values (S3 Table).

For the genus *Paenarthrobacter*, we compared the complete genomes of GOM3 strain and of *Paenarthrobacter nicotinovorans* ATCC 49919, which was identified as the closest related defined species to ours based on TYGS analysis (S2 Table). A comparison of these genomes revealed that they shared 7 out of 9 total syntenic blocks, covering only 53.56% of GOM3 and 53.55% of ATCC 49919 (Fig 2C). Our observation of this lower synteny conservation supports the notion that factors like HGT or adaptation to specific ecological niches may have played a role in the evolutionary divergence that led to the speciation of GOM3. In the case of the marine strain *P. aeruginosa* GOM9 and the nosocomial strain *P. aeruginosa* DSM 50071, 14 out of 15 aligned blocks were shared between their genomes, with over 96% coverage (Fig 2D).

## KEGG functional predictions and secondary metabolite gene clusters found in the genomes from the marine consortium

To examine general functional distinctions between the consortium strains, gene annotation was performed using the KEGG functional database (Fig 3A). A higher abundance of genes associated with 'carbohydrate metabolism', 'amino acid metabolism', and 'cellular processes' was observed across the genomes (Fig 3A, first panel). The KO numbers for each genome are provided in S4 Table.

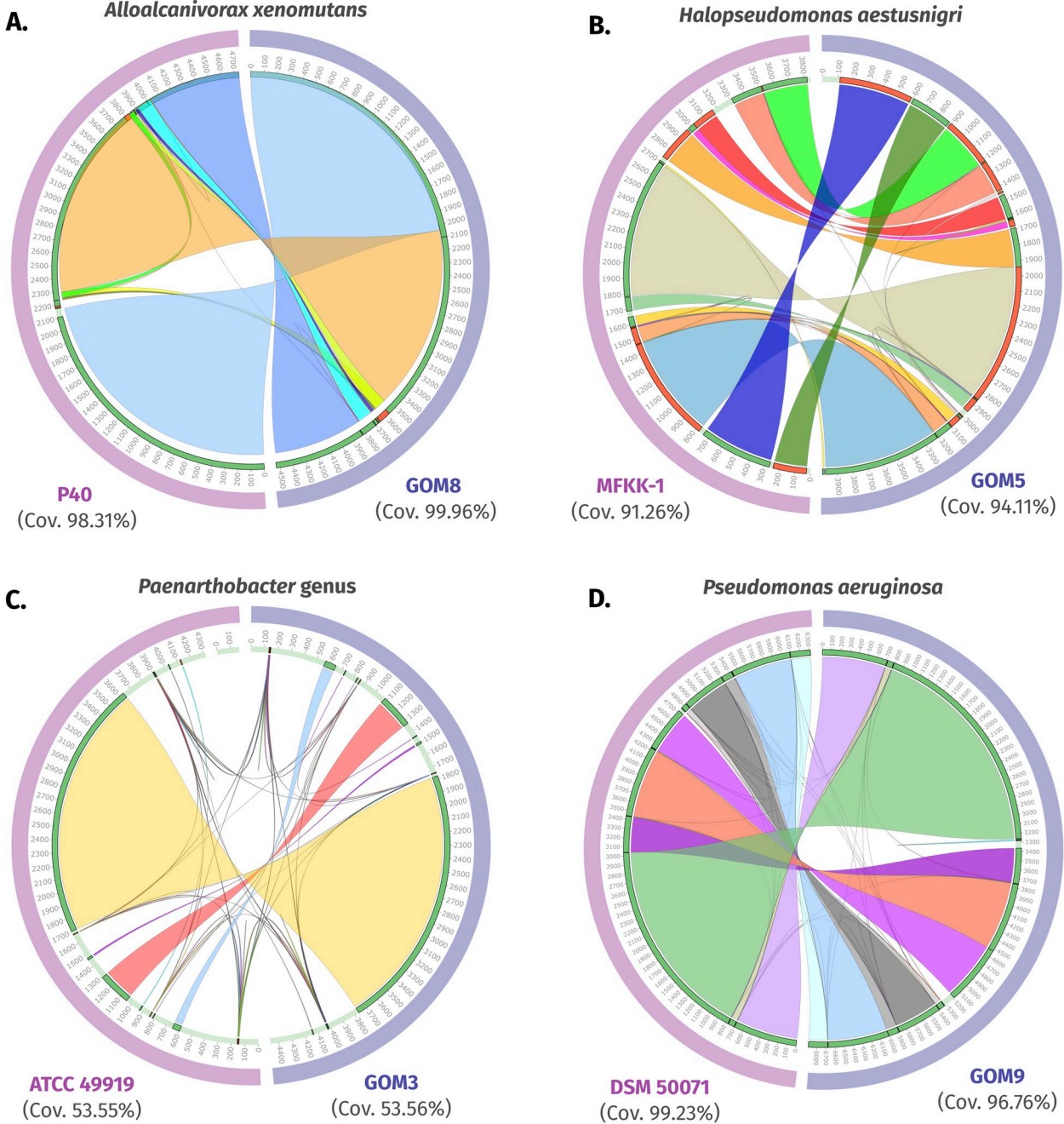

**Fig 2. Comparison of syntenic blocks between consortium genomes and complete reference genomes using the Sibelia tool with default parameters.** (A) *Alloalcanivorax xenomutans* GOM8 vs. *A. xenomutans* P40. (B) *Halopseudomonas aestusnigri* GOM5 vs. *Halopseudomonas* sp. MFKK-1. (C) *Paenarthrobacter* sp. GOM3 vs. *Paenarthrobacter nicotinovorans* ATCC 49919. (D) *Pseudomonas aeruginosa* GOM9 vs. *P. aeruginosa* DSM 50071. From the outermost ring to the center, the content is as follows: Each pair of genomes is marked in blue and purple, the size of each genome is indicated with gray numbers, the contigs of each genome are marked in pale green (darker green contigs indicate a positive direction, while red indicates a negative direction of the sequences), and different colors represent the syntenic blocks.

The *Paenarthrobacter* sp. GOM3 genome differed from the other three consortium genomes due to its distinct phylum affiliation, and presented variance in carbohydrate metabolism (Fig 3A, second panel). *P. aeruginosa* GOM9 had the highest number of nonshared

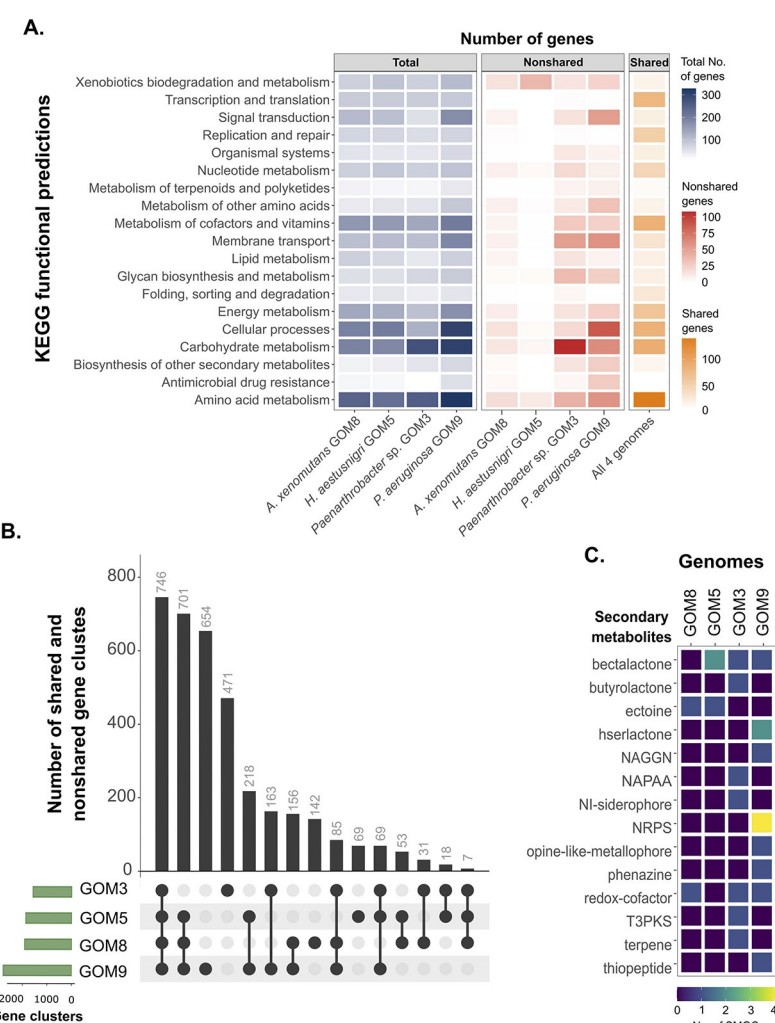

**Fig 3. KEGG functional and secondary metabolite-related gene cluster (SMGC) predictions for the consortium bacterial genomes.** (A) Comparison between the KO markers on level 2 of the KEGG functional category. The total gene numbers are in blue, unique or nonshared KOs in red, and the shared KO counts or the intersection for all four genomes in orange. The more intense colors observed in the *P. aeruginosa* GOM9 genome in the first panel, in contrast to the other three, are consistent with its larger genome size. (B) UpSet plot with the number of shared and nonshared KOs. (C) Heatmap with the SMGCs predicted with the antiSMASH software in 'strict' mode.

genes in cellular processes, some of which are related to biofilm formation. *P. aeruginosa* is a well-known biofilm former and HC degrader [10,33]. Studies have shown that biofilm formation plays a key role in the degradation of aromatic HCs. Biofilms provide a favorable microenvironment for these bacteria to thrive and efficiently break down these complex molecules [33].

The number of nonshared genes involved in 'xenobiotics biodegradation and metabolism' was significantly higher in the *H. aestusnigri* GOM5 assembly. This functional category encompasses the degradation of certain HCs, including benzoate and aminobenzoate, as well as drug metabolism. Detailed gene annotations related to HC degradation will be discussed in the subsequent section. The most prevalent functional categories shared by all four genomes included 'amino acid metabolism', 'carbohydrate metabolism', 'metabolism of cofactors and vitamins', 'cellular processes', and 'transcription and translation' (Fig 3A, third panel).

Utilizing the predicted KO numbers as ortholog gene clusters, comparative analysis revealed a total of 3,583 clusters across the four strains (Fig 3B). Of these, 746 clusters were common to all four genomes, while 701 clusters were shared by the three Gammaproteobacteria strains. Additionally, 1,336 gene clusters were exclusive to a single genome, with *Pseudomonas* having 654, *Paenarthrobacter* having 471, *Alloalcanivorax* having 142, and *Halopseudomonas* having 69. The highest number of clusters shared between at least two genomes was observed between *Halopseudomonas* and *Pseudomonas* with 218, followed by *Paenarthrobacter* and *Pseudomonas* with 163.

Regarding the secondary metabolite-related gene clusters (SMGCs) predicted by antiSMASH, *P. aeruginosa* GOM9 had the highest number of clusters (12), including four identical to *P. aeruginosa* PAO1 clusters (S5 Table). These SMGCs include the biosynthesis of the peptide azetidonamide A (BGC0002037) regulated by quorum sensing [34], the toxin and antibiotic L-2-amino-4-methoxy-trans-3-butenoic acid (BGC0000287), the antimicrobial pigment pyocyanin (BGC0000936) [35], and the metallophore pseudopaline (BGC0002489) [36]. *Paenarthrobacter* sp. GOM3 had seven SMGCs, one of which matched the desferrioxamine E cluster (BGC0001478) of *Streptomyces* sp. ID38640 [37], a clinical siderophore for treating iron overload in humans. *A. xenomutans* GOM8 and *H. aestusnigri* GOM5 had two and three SMGCs, respectively, of betalactone, ectoine, and redox-cofactor types. However, none of them had 100% similarity with known clusters. All genomes contained approximately 67% SMGCs with <31% similarity to known clusters, totaling sixteen, indicating the potential for new natural products or antimicrobials from these marine bacteria.

## Presence and refined annotation of HC-degrading genes

Genes involved in HC aerobic degradation were predicted for the four marine consortium strains (Fig 4). *P. aeruginosa* GOM9 exhibited the highest number of genes: 22 for alkane degradation, 31 for aromatic degradation, and 5 for biosurfactant production. This result likely explains the faster light crude oil degradation observed in individual cultures of *P. aeruginosa* GOM9 during the first seven days of incubation [15]. *Paenarthrobacter* sp. GOM3 had the second-highest gene count, with 4 genes for alkane degradation and 30 for aromatics. *A.*

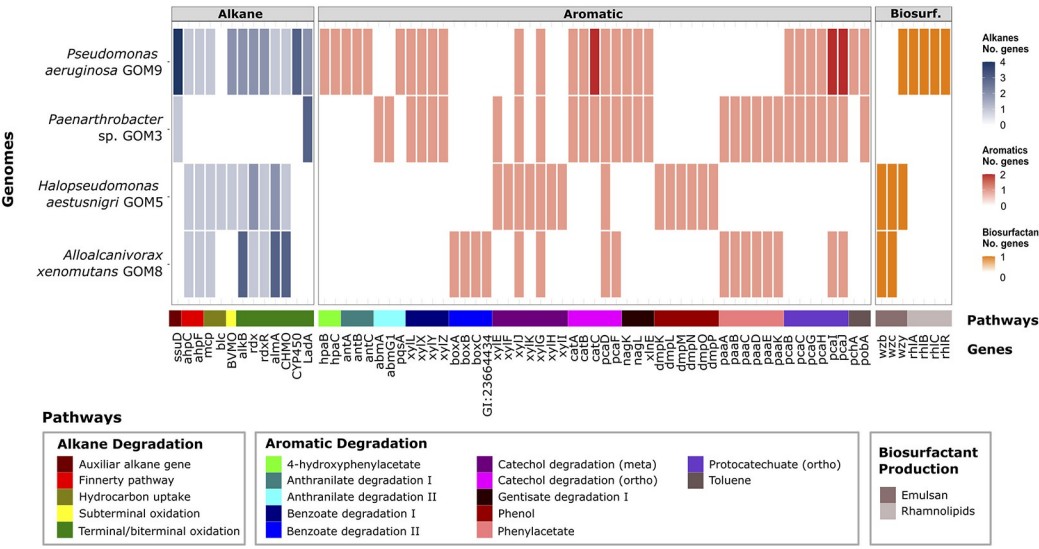

**Fig 4. Genes related to the aerobic HC degradation of alkanes and aromatics.** The functional prediction was made using the HADEG database and the RAST-tk annotation.

*xenomutans* GOM8 followed with 14 for alkane degradation, 16 for aromatics, and 2 for biosurfactants. *H. aestusnigri* GOM5 had 12 genes for alkane degradation, 14 for aromatics, and 3 for biosurfactant production.

Interestingly, all four strains shared only three genes, *xylJ*, *xylG* and *pcaD*, which encode enzymes (2-oxopent-4-enoate hydratase, 2-hydroxymuconate semialdehyde dehydrogenase, and 3-oxoadipate enol-lactonase, respectively) involved in the meta and ortho-cleavage pathways of catechol (Fig 4). This metabolic complementarity potentially explain why the consortium achieved a higher light crude oil removal (62%) compared to any individual strain (maximum 39%) [15]. Microbial consortia create a platform to divide the labor of metabolic pathways [38]. In our case, the consortium offered broader potential metabolic pathway coverage for HC degradation than any single strain could achieve alone.

## Alkane degradation potential

The genomes of all three proteobacteria had genes associated with the four alkane metabolic pathways: the terminal oxidation pathway, the biterminal pathway, the subterminal pathway, and the Finnerty pathway (Fig 4). Among these, *A. xenomutans* GOM8 had the highest number of *alkB* and *almA* genes (three each), which encode proteins that introduce oxygen at the terminal carbon of alkanes, leading to alcohols. Additionally, GOM8 had the highest number of cyclohexanone monooxygenase (CHMO) genes (three copies), essential for degrading cycloaliphatic HCs [39]. *Alloalcanivorax* genus specializes in degrading aliphatic HCs [4,40,41]. This specialization was evident across the 142 genomes used for phylogenetic reconstruction (S1 Fig). Interestingly, all *Alloalcanivorax* genomes within the dataset exhibited the highest counts of *alkB*, *almA*, and CHMO genes.

The genome of *P. aeruginosa* GOM9 contained several genes related to alkane degradation including *alkB*, *almA*, CHMO, and Baeyer-Villiger monooxygenase (BVMO) (Fig 4). Notably, the GOM9 strain was the only strain in the consortium to possess a P450 cytochrome, which plays a key role in degrading various alkanes and aromatic compounds [42–44]. In addition, both *P. aeruginosa* GOM9 and *Paenarthrobacter* sp. GOM3 were the only strains harboring *LadA*, and *ssuD* genes, associated with alkane terminal oxidation [45] and alkanesulfonates degradation [46], respectively. These genes were unique to GOM3 regarding alkane degradation, suggesting its potential involvement in degrading other HCs, as previously suggested [9]. Among the 61 annotated *Pseudomonas* genomes (S1 Fig), the *P. aeruginosa* subclade showed the highest number of alkane degradation-related genes, indicating its potential superior efficiency in utilizing alkanes compared to other *Pseudomonas* species, regardless of whether they were environmental or nosocomial isolates. Remarkably, within the *Pseudomonas* subclade, GOM1 and GOM9 strains from the GoM exhibited an additional rubredoxin-NAD(+) reductase (*rdxR*), crucial for electron transfer to *alkB*. Furthermore, the GOM1 genome contained an extra *almA* gene. These extra genes suggest adaptation events in the HC-contaminated marine environment from which these strains were isolated [5,15].

Regarding the *H. aestusnigri* GOM5 strain, the presence of genes such as *ahpCF*, *alkB*, *almA*, and CHMO in its genome suggests its potential capability to degrade alkanes via the Finnerty and terminal oxidation pathways. A comparative analysis among *Halopseudomonas* genomes revealed that *H. aestusnigri* genomes, along with those of seven other species, harbor at least one BVMO gene related to the subterminal oxidation pathway (S1 Fig).

## Aromatic degradation potential

Aromatic-degradation genes displayed redundancy within the consortium, but also there were unique and complete gene clusters in specific genomes (Fig 4). In *P. aeruginosa* GOM9,

harbored unique genes for degrading 4-hydroxyphenylacetate (*hpaBC*) and converting anthranilate to catechol (*antABC*). Shared pathways between *P. aeruginosa* GOM9 and *Paenarthrobacter* sp. GOM3, such as benzoate degradation pathway I (*xylXYZL*), and ortho pathways for catechol (*catABC*, *pcaDF*), and protocatechuate degradation (*pcaBCGHIJ*), suggest potential competition for these carbon sources in the consortium. Furthermore, *Paenarthrobacter* had unique genes for anthranilate degradation pathway II (*abmA*, *abmG1*), and shared phenylacetate degradation genes (*paaABCDEK*) with *A. xenomutans* GOM8. *A. xenomutans* GOM8 featured a distinct gene cluster for benzoate degradation pathway II (*boxABC*) involving benzoyl-CoA formation. Finally, *H. aestusnigri* GOM5 uniquely possessed a complete cluster for phenol degradation (*dmpKLMNOP*) and meta-pathway catechol degradation (*xylEFJKGHI*).

Some of these genes encoded proteins with broad substrate specificity. For instance, the *xylXYZ* complex in *P. putida* mt-2 and *Acinetobacter calcoaceticus* ADP1 can process various substituted benzoates [47]. This complex in *P. putida* mt-2 shared 72.10%-77.78% and 52.55%-61.73% sequence identity with the *xylXYZ* of *P. aeruginosa* GOM9 and *Paenarthrobacter* sp. GOM3, respectively. Similarly, pyrocatechase or catechol 1,2-dioxygenase (*catA*) can catalyze reactions with diverse substrates, such as phenol, 2,4-dichlorophenol, or 4-nitrophenol in *P. chlororaphis* UFB2 [48] and pyrogallol and hydroxyquinol in *Blastobotrys raffinosifermentans* [49]. The amino acid sequences of both microorganisms shared identities ranging from 27.15% to 30.25% with *catA* in GOM3 and 30.74% to 72.17% with *catA* in GOM9. This feature of some aromatic degrading proteins extends the range of potential substrates for the consortium bacteria beyond those explicitly mentioned in pathway names (Fig 4).

The comparative analysis of the 142 genomes revealed that among the set of 32 *Paenarthrobacter* genomes, only *Paenarthrobacter* sp. GOM3 contained both the *xylLXYZ* and *catABC* genes (S1 Fig), supporting previous findings [16]. A BLASTP analysis (PSI-BLAST) of the benzoate 1,2-dioxygenase alpha subunit sequence (*xylX*) and the catechol 1,2-dioxygenase (*catA*) sequences against the NCBI nonredundant database confirmed the previously obtained result (consulted on September 1, 2023). Notably, none of the 500 sequences closest to both enzymes belonged to the genus *Paenarthrobacter*. Instead, the most closely related sequence to *xylX* was from *Pseudarthrobacter* sp. NIBRBAC000502771 (98.92% identity, 100% coverage), while the nearest relative to our *catA* was *Arthrobacter* sp. 8AJ (95% identity, 97% coverage). The close relationship between the genera *Paenarthrobacter* and *Arthrobacter* suggests that our GOM3 strain may have gained (through HGT) or retained these dioxygenases during its evolution, and they may play an adaptive role in the environment (marine sediment from the GoM) in which this strain was isolated.

Of the 142 analyzed genomes, only *A. xenomutans*, *A. balearicus*, and *A. dieselolei* possessed the benzoate degradation II or box pathway (S1 Fig). This pathway is active under low-oxygen conditions and has been identified in several Proteobacteria genera [50]. The box gene cluster (*boxABC*) has been proposed as a genetic marker for monitoring HC pollution in marine environments [51]. The PSI-BLAST analysis of benzoate-CoA ligase (GI:23664434 or *bclA*) and benzoyl-CoA oxygenase component A (*boxA*) from *A. xenomutans* GOM8 revealed that sequences from *A. balearicus* MACL04, *A. dieselolei* B5, *Alcanivorax* sp. 24, and *Alcanivorax xiamenensis* 6-D-6 had the highest similarities (identities > 93%, coverage 100%). It can be inferred that these microorganisms have adjusted to conditions with limited oxygen. To date, there have been no experimental reports of this pathway described within the genera *Alcanivorax* or *Alloalcanivorax*.

Comparing aromatic genes across the 24 *Halopseudomonas* genomes, diverse pathways were identified. For instance, the *H. gallaeciensis*, *H. pachastrellae*, *H. pelagia*, and *H. aestusnigri* genomes exclusively possess genes for catechol degradation via the meta pathway (*xylEFJKGHI*) and phenol (*dmpKLMNOP*). Additionally, *H. bauzanensis*, *H. litorales*, *H. pertucinogena*,

and some *Halopseudomonas* sp. harbor gene sets for benzoate degradation I (*benAB*, *xylZL*) and catechol via the ortho pathway (*catABC*, *pcaDF*). These distinctions may arise from bacterial environmental adaptations, a phenomenon widely observed in the bacterial world [52].

## Biosurfactant production potential

Regarding biosurfactant production, crucial for enhancing the bioavailability of petroleum HCs [53], three consortium genomes contained genes for emulsan or rhamnolipid biosurfactants (Fig 4). *A. xenomutans* GOM8 had two genes related to emulsan synthesis (*wzb*, *wzc*) but lacked *wzy*, similar to the other 17 *Alloalcanivorax* assemblies (S1 Fig). To the best of our knowledge, emulsan production has not been reported for this species or genus. A closely related strain, *A. dieselolei* B5, has been shown to produce a proline-based biosurfactant when grown with diesel oil [54]. Our four genomes harbored genes for proline synthesis, suggesting the consortium strains might also be capable of producing this type of biosurfactant. Proline plays a crucial role in adaptation to osmotic and dehydration stresses, redox control, and apoptosis [55,56].

The genome of *H. aestusnigri* GOM5 harbored three genes related to emulsan synthesis (*wzb*, *wzc*, *wzy)* (Fig 4), indicating potential emulsan production. While bacteria belonging to the *H. pertucinogena* lineage, which includes *H. aestusnigri*, have been linked to emulsan production [57], our annotation identified an additional wzy gene not previously reported.

The presence of *rhlABCR* genes in *P. aeruginosa* GOM9 suggests its ability to synthesize mono- and di-rhamnolipids synthesis, potentially improving HC access for the consortium [15]. HC degradation enhancement due to biosurfactant was previously reported in a mixed culture of *Acinetobacter* sp. XM-02 and *Pseudomonas* sp. XM-01 [13]. The XM-01 strain produced rhamnolipids that reduced the crude oil's surface tension, leading to enhanced HC degradation compared to cultures containing each strain individually.

The genomes of *Paenarthrobacter* lacked genes associated with HADEG biosurfactant (S1 Fig), suggesting they might produce alternative types, possibly including the previously mentioned proline-based biosurfactant. The *Bacillus* genomes, used as an outgroup, contain complete gene sets for iturin, plipastatin, and surfactin, as expected [58]. *Bacillus* species are known to utilize biosurfactants as virulence factors for defense purposes [59].

## Hydrocarbon-related gene gain and loss in the consortium and their relatives

To predict the gain and loss of orthologous genes, an ancestral reconstruction analysis was conducted on the 142 genomes using COUNT software and the orthologous groups identified by Proteinortho (Fig 5). The results revealed a higher number of gene gain events than gene loss events (Fig 5A). Among the predicted common ancestors for each genus, the *Paenarthrobacter* common ancestor (PAECA) had 2,912 protein families (PrFs), representing approximately 72% of the average PrFs in the selected *Paenarthrobacter* genomes (S6 Table). The *Alloalcanivorax* common ancestor (ACA) shared 2,413 PrFs with *Alloalcanivorax* assemblies, corresponding to approximately 62% of their average PrFs. The *Halopseudomonas* common ancestor (HCA) shared 1,991 PrFs with *Halopseudomonas* assemblies, equivalent to approximately 59% of their average PrFs. Last, the *Pseudomonas* common ancestor (PCA) shared 2,320 PrFs with the selected *Pseudomonas* genomes, constituting approximately 44% of their average. In general, the highest percentage of the PrFs in PAECA suggests a lower number of gained genes among *Paenarthrobacter* species compared with the other three genera.

The analysis further showed that all genomes in each subclade of the consortium strains at the species level shared over 88% of their PrFs (S6 Table), with the highest percentage shared

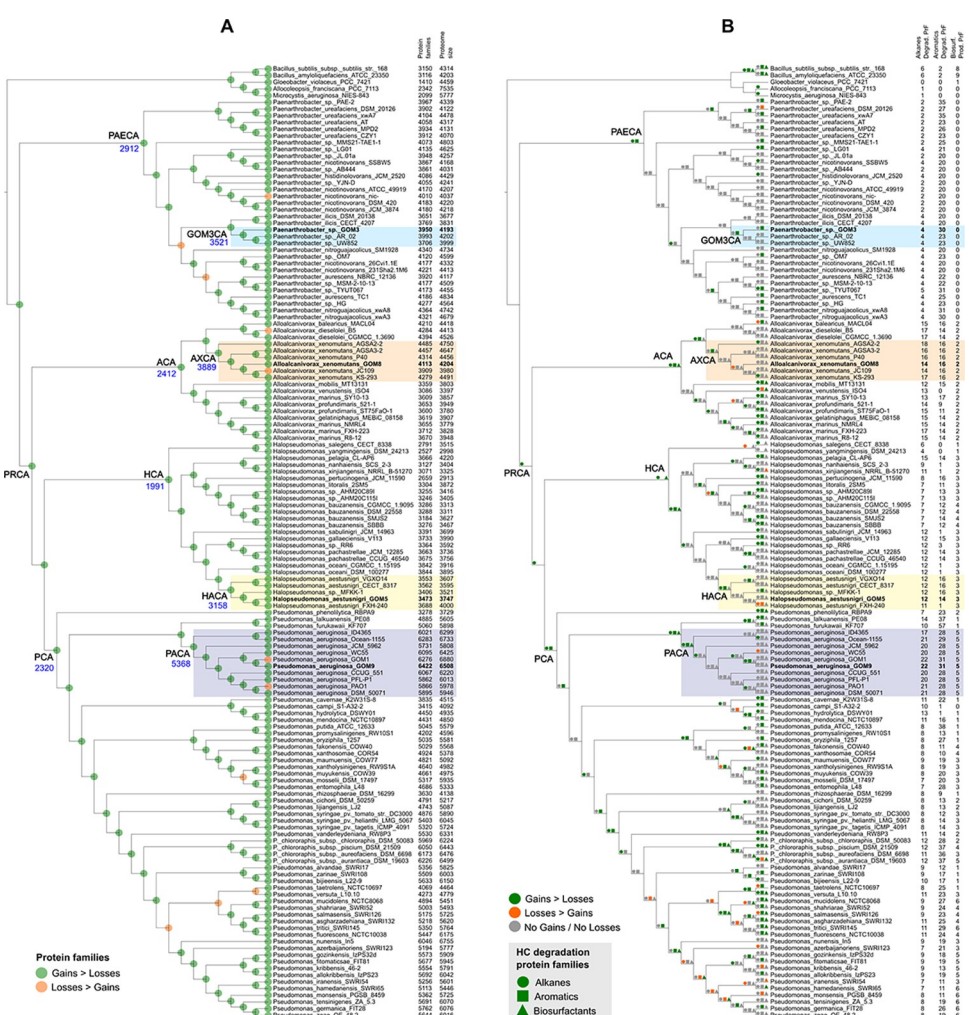

**Fig 5. Gain and loss of orthologous genes in the consortium and their relatives of the genomes selected for phylogenomic inference.** (A) Total protein families. The circles represent the gain (green), losses (orange), and no gains/losses (gray) in each node. (B) Protein families related to HC degradation and biosurfactant production. The HCs are shown as circles (alkanes) and squares (aromatics), and the biosurfactants are shown as triangles.

among the *A. xenomutans* genomes (approximately 91%), whose common ancestor (AXCA) had 3,889 PrFs (Fig 5A). This suggests that less than 12% of the PrFs were gained by these strains.

Our analysis revealed that PrFs related to alkane degradation in *Paenarthrobacter* sp. GOM3 genome could be traced back to two of its ancestors, PAECA, and two nodes prior to the common ancestor of GOM3 (GOM3CA) (Fig 5B). Moreover, some of the PrFs associated with aromatic degradation could be traced back to PAECA and GOM3CA. However, PrFs related to the *xylLYXZ* and *catABC* genes lacked traceable ancestors on the tree, implying recent acquisition. Likewise, *Paenarthrobacter* sp. PAE-2, a strain isolated from waste-contaminated soil, is likely to have recently gained aromatic-degradation PrFs related to genes for phthalate and protocatechuate (meta) degradation (S1 Fig). Notably, this strain possessed the highest number of genes related to aromatic degradation among the *Paenarthrobacter* strains.

The remaining three consortium genomes likely inherited PrFs linked to HC degradation and biosurfactant production from their predicted ancestors. For instance, some PrFs

associated with alkane degradation were traced back to the common ancestor of the analyzed Proteobacteria genomes (PRCA) (Fig 5B). These PrFs are related to the rubredoxin gene (*rdx*) and a gene encoding a methyl-accepting chemotaxis protein (*alkN* or *mcp*) responsible for HC uptake. Furthermore, the analysis suggests that the immediate ancestor to *A. xenomutans* GOM8 lost a PrF (Fig 5B) related to the CYP450 gene, which was gained by ACA but absent in the GOM8 and JC109 assemblies (Fig 5). Furthermore, the KS-293 strain exhibited a regression to the ACA ancestral state.

The ancestral reconstruction predicted that the common ancestor of *H. aestusnigri* (HACA) gained phenol degradation genes (*dmpKLMNOP*), which were possibly transmitted as orthologs to the genomes of the same species, except for the FXH-240 isolate. Furthermore, the common ancestor of *P. aeruginosa* (PACA) acquired the genes for rhamnolipid production and some associated with alkane and aromatic degradation, passing them vertically to strains within this species.

## Further directions

Our study of this consortium establishes a foundation for its potential application in bioremediation strategies, such as *in situ* bioaugmentation. The results from the well-controlled laboratory experiments [15] combined with this current genomic analysis provide valuable insights into the consortium's capabilities. To translate these findings from the lab environment to real-world scenarios, further investigations are necessary. Scaling-up experiments using mesocosms (which introduce a more complex and semi-controlled environment) [60,61], testing different inoculation dosage strategies [62], or assessing the efficiency of using a lyophilized consortium [63] could be potential next steps.

Building on the current genomic findings, future research employing a global transcriptional profile of the consortium would be valuable. This approach would provide a more comprehensive understanding of the consortium's degradative capabilities by revealing which of the identified genes are actively expressed during HC degradation. Regarding the presence of genes related to potential biosurfactant production, future activity-based assays (e.g., droplet collapse assay, surface tension measurement) and potentially isolating/purifying the biosurfactants would provide more characterization of these strains' capabilities.

## Conclusions

We conducted a comparative analysis of high-quality circular genomes from the four marine bacteria within an HC-degrading consortium. Taxonomically, we identified strains GOM8 and GOM9 as *A. xenomutans* and *P. aeruginosa*, respectively, that were lacking species-level taxonomy. Strain GOM5 was reconfirmed as *H. aestusnigri*, while strain GOM3 was verified as a novel species belonging to the *Paenarthrobacter* genus.

The functional annotation analysis revealed potential metabolic differences among the four bacteria, highlighting their diverse metabolic capabilities and potential for the development of new natural products and antimicrobials. Exploring these aspects could be the subject of future research endeavors. The analysis of HC-degrading genes and biosurfactant production genes also highlights the functional diversity of these bacteria, suggesting competition for resources and specialization in utilizing certain substrates. The metabolic complementarity within the consortium likely explains its enhanced light crude oil removal efficiency compared to individual strains. A further transcriptomic study would be valuable to reveal which of these genes are actively expressed during HC exposure.

Our gain and loss analysis provides valuable insight into the evolutionary history of marine consortium genomes. The results suggest that while some HC degradation and biosurfactant

production genes have deep ancestral origins, others may have been acquired more recently, reflecting the dynamic nature of microbial adaptations. Notably, *Paenarthrobacter* sp. GOM3 seems to have recently gained genes related to aromatic degradation, which could increase its fitness under its specific environment.

These findings significantly enhance our understanding of the genomic characteristics, taxonomic identities, and evolution of these marine bacteria. They shed light on their ecological roles and potential biotechnological application, particularly in HC degradation and the development of environmental remediation strategies.

## Supporting information

**S1 Fig. Genes related to the aerobic HC degradation of alkanes and aromatics.** The functional prediction was made using the HADEG database and the RAST-tk annotation.
(PDF)

**S1 Table. List of the 138 genomes used for phylogenomic reconstruction.** The source information was retrieved from NCBI consulted on July 5, 2023. The metadata was obtained from the GTDB database (https://gtdb.ecogenomic.org/) consulted on June 30, 2023.
(XLSX)

**S2 Table. Phylogenomic taxonomic comparison of each strain of the marine consortium with type strain prokaryotic genomes included in TYGS.**
(XLSX)

**S3 Table. Percentage values of ANIm and [aligned nucleotides] between the consortium genomes and the assemblies within the subclades from the phylogenetic tree (Fig 1).**
(XLSX)

**S4 Table. KO numbers of each marine bacteria.**
(XLSX)

**S5 Table. Identified secondary metabolite regions of the consortium genomes using anti-SMASH v7.0.1 with strictness 'strict'.**
(XLSX)

**S6 Table. The number of protein families of predicted common ancestors.**
(XLSX)

## Acknowledgments

We thank Jerome Verleyen for technical support and access to HPC infrastructure at the Unidad Universitaria de Secuenciación Masiva y Bioinformática, Instituto de Biotecnología (UNAM) which is part of the Laboratorio Nacional de Apoyo Tecnológico a las Ciencias Genómicas (CONAHCyT). We thank MC Karla Sofía Millán-López and MC Itzel Hidalgo-Manzano for their technical support.

## Author Contributions

**Conceptualization:** Jorge Rojas-Vargas, Liliana Pardo-López.

**Data curation:** Jorge Rojas-Vargas.

**Formal analysis:** Jorge Rojas-Vargas.

**Funding acquisition:** Liliana Pardo-López.

**Methodology:** Jorge Rojas-Vargas, Alejandro Sanchez-Flores.

**Project administration:** Liliana Pardo-López.

**Resources:** Liliana Pardo-López.

**Software:** Jorge Rojas-Vargas.

**Supervision:** Liliana Pardo-López.

**Validation:** Eria A. Rebollar, Alejandro Sanchez-Flores.

**Visualization:** Jorge Rojas-Vargas.

**Writing – original draft:** Jorge Rojas-Vargas.

**Writing – review & editing:** Eria A. Rebollar, Alejandro Sanchez-Flores, Liliana Pardo-López.

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
