## [Decision Letter · Decision Letter 0]

4 Mar 2024

PONE-D-23-38809A comparative genomic study of a hydrocarbon-degrading marine bacterial consortiumPLOS ONE

Dear Dr. Pardo-López,

Thank you for submitting your manuscript to PLOS ONE. After careful consideration, we feel that it has merit but does not fully meet PLOS ONE’s publication criteria as it currently stands. Therefore, we invite you to submit a revised version of the manuscript that addresses the points raised during the review process.

We look forward to receiving your revised manuscript.

Kind regards,

Pankaj Kumar Arora

Academic Editor

PLOS ONE

Journal Requirements:

"This work was supported by UNAM-PAPIIT IG200223 and CONACYT – Mexican Ministry of Energy- Hydrocarbon Trust, project 201441." 

Reviewers' comments:

Reviewer's Responses to Questions

**Comments to the Author**

1. Is the manuscript technically sound, and do the data support the conclusions?

Reviewer #1: Yes

Reviewer #2: Yes

2. Has the statistical analysis been performed appropriately and rigorously? 

Reviewer #1: Yes

Reviewer #2: Yes

3. Have the authors made all data underlying the findings in their manuscript fully available?

Reviewer #1: Yes

Reviewer #2: Yes

4. Is the manuscript presented in an intelligible fashion and written in standard English?

Reviewer #1: Yes

Reviewer #2: Yes

5. Review Comments to the Author

Reviewer #1: PONE-D-23-38809

In this manuscript, a study on a bacterial community isolated from an hydrocarbons contaminated marine site was conducted. The genome of four isolates was investigated in detail and a plethora of information was obtained, focusing on identification of genes encoding for proteins involved in hydocarbons degradation, allowing to shed light on the role isolated strains may have in oil hydrocarbons degradation and giving a base for possible bioremediation uses.

This study offers an interesting view of the role different members of a microbial consortium may perform, thus conducting to an efficient degradation of an organic complex mixture. Oil hydrocarbons with their aliphatic and aromatic components, their hydrophobic nature, their scarce bioavailability, can be used as carbon and energy source by heterotrophic bacteria.

The manuscript is well written and can give insights for future research.

Revisions

Line 25: ‘… 138 bacterial species …’ change to ‘… 138 bacterial stains …’;

Biosurfactants allow better bioavailability of hydrophibic substrata as hydrocarbons, in the meantime, adhesion of bacterial cells to hydrocarbons can represent an important aspect favoring bacterial degradation of hydrophibic substrata. Investigation in this context could add information on the efficiency of these isolates in degrading hydrocarbons in bioremediation processes.

Reviewer #2: 1 The manuscript is well written; however, the author needs to address following comments

2. Improve the keywords, that should be reflected the nature of the work

3. Introduction part needs enhancement.

4. What are the potential sources of contamination, and how was this contamination assessed?

5. Check the typological error and language quality.

6. What was the specific methodology used to isolate the bacterial strains or give any references for isolation.

7. Rewrite the Consortium strains and nucleic acid extraction

8. How to conform the biosurfactant production of this bacterial strain (any specific analysis).

9. The images are not clear give high-DPI quality images.

10. Check the References are placed correctly.

6. PLOS authors have the option to publish the peer review history of their article (what does this mean?). If published, this will include your full peer review and any attached files.

Reviewer #1: No

Reviewer #2: **Yes: **Aruliah rajasekar

---

## [Author Response · Author response to Decision Letter 0]

13 Mar 2024

RESPONSE TO REVIEWERS

Please find below the responses to the comments from the Reviewers:

Reviewer #1

In this manuscript, a study on a bacterial community isolated from a hydrocarbon contaminated marine site was conducted. The genome of four isolates was investigated in detail and a plethora of information was obtained, focusing on identification of genes encoding for proteins involved in hydocarbons degradation, allowing to shed light on the role isolated strains may have in oil hydrocarbons degradation and giving a base for possible bioremediation uses.

This study offers an interesting view of the role different members of a microbial consortium may perform, thus conducting to an efficient degradation of an organic complex mixture. Oil hydrocarbons with their aliphatic and aromatic components, their hydrophobic nature, their scarce bioavailability, can be used as carbon and energy source by heterotrophic bacteria.

The manuscript is well written and can give insights for future research.

Line 25: ‘… 138 bacterial species …’ change to ‘… 138 bacterial stains …’

Thank you for your observation. L25, now L26, has been modified by changing "species" to "strains."

Biosurfactants allow better bioavailability of hydrophibic substrata as hydrocarbons, in the meantime, adhesion of bacterial cells to hydrocarbons can represent an important aspect favoring bacterial degradation of hydrophibic substrata. Investigation in this context could add information on the efficiency of these isolates in degrading hydrocarbons in bioremediation processes.

Reviewer #2

1. The manuscript is well written; however, the author needs to address the following comments

2. Improve the keywords, that should be reflected the nature of the work

Thank you for the comment. The new keywords in L38-39 are “marine bacterial consortium, hydrocarbon-degradation genes, functional diversity, comparative genomics, pangenome analysis”.

3. Introduction part needs enhancement.

The introduction was improved emphasizing the environmental impact of oil spills (L41-43). It also tightens the connection between HC pollution and bioremediation, focusing on the role of bacteria as a natural solution (L44-48). Additionally, complex sentences have been rephrased and redundant wording removed, improving overall readability (L60-80).

4. What are the potential sources of contamination, and how was this contamination assessed?

Regarding the first question, the new Introduction highlights in L41: "Crude oil spills, a consequence of anthropogenic activities in the ocean" acknowledging the human role in these environmental disasters. 

For the second question, oil spill contamination can be assessed through different methods like visual observation, chemical analysis of water and sediment samples, and remote sensing (NOAA 2020; Mauseth and Parker 2011). Our present research focuses on the bioremediation stage that follows the initial assessment. By understanding the genetic content of HC-degrading bacteria, we can develop more effective strategies to utilize these natural clean-up crews for future spills.

References

Mauseth, Gary S., and Heather Parker. 2011. “Natural Resource Damage Assessment.” In Oil Spill Science and Technology, 1067–82. Elsevier.

NOAA. 2020. “Oil Spills.” August 1, 2020. https://www.noaa.gov/education/resource-collections/ocean-coasts/oil-spills.

5. Check the typological error and language quality.

Thank you for your observation, the text was revised again, and we believe the errors were corrected.

6. What was the specific methodology used to isolate the bacterial strains or give any references for isolation.

Thank you for your question regarding the methodology used to isolate the bacterial strains. We apologize for not including this information in the previous draft. We have added a section under "Consortium strains and nucleic acid extraction" detailing the isolation process (L84-88). Briefly, the strains were isolated from the Gulf of Mexico using enrichment culture techniques followed by purification through streaking [reference the isolation method in reference 15]. We hope this clarifies our methodology.

7. Rewrite the Consortium strains and nucleic acid extraction

 Done as we described in the previous question.

8. How to conform the biosurfactant production of this bacterial strain (any specific analysis).

While our study focused on predicting the metabolic potential for hydrocarbon degradation and biosurfactant production, confirming biosurfactant production through specific analyses (e.g., droplet collapse assay, and surface tension measurement) would be a valuable addition to future research but was not part of the main scope of this work. We added this suggestion in L485-488.

9. The images are not clear give high-DPI quality images.

Done. The new version has all images in high quality according to the PACE tool by PLoS.

10. Check the References are placed correctly.

The text has been reviewed, and we believe the References are placed correctly.

---

## [Editor Report · Decision Letter 1]

20 Mar 2024

PONE-D-23-38809R1A comparative genomic study of a hydrocarbon-degrading marine bacterial consortiumPLOS ONE

Dear Dr. Pardo-López,

Thank you for submitting your manuscript to PLOS ONE. After careful consideration, we feel that it has merit but does not fully meet PLOS ONE’s publication criteria as it currently stands. Therefore, we invite you to submit a revised version of the manuscript that addresses the points raised during the review process.

We look forward to receiving your revised manuscript.

Kind regards,

Pankaj Kumar Arora, Ph.D

Academic Editor

PLOS ONE

Journal Requirements:

Reviewers' comments:

**Reviewer#3:**

The manuscript entitled “A comparative genomic study of a hydrocarbon-degrading marine bacterial consortium” by Pardo-López et al. presents a comparative analysis of four bacterial genomes. The marine bacteria were isolated in the Gold of Mexico in a previous paper published by the group (Rojas-Vargas J, Adaya L, Silva-Jiménez H, Licea-Navarro AF, Sanchez-Flores A, Gracia A, et al. Oil-degrading bacterial consortium from Gulf of Mexico designed by a factorial method, reveals stable population dynamics. Frontiers in Marine Science. 2022;9. doi:10.3389/fmars.2022.962071). Bacteria were used in consortium to degrade hydrocarbons in the previous report with success. In this paper the group presents the sequence of the bacterial genomes and analysis. The genome sequence of the 4 isolates allowed 3 important conclusions: (i) a taxonomic study and a phylogenetic reconstruction; (ii) the genome sequences allowed the identification of the genes related to alkane, aromatic and biosurfactant biosynthetic pathways in each isolate; and (iii) the gain and loss of the genes involved in hydrocarbon metabolism. From the knowledge and analysis of the sequences the authors propose complementary metabolic functions for the bacteria which would explain the higher hydrocarbon assimilation in the consortium compared to the individual bacterium.

I find it important to gain knowledge of the genomes in a technical way, however, it would be much more interesting if the authors would have done a global transcriptional profile of the bacterial consortium, since the mere presence of the genes in the genomes do not guarantee its expression and function. The authors would have a much more precise and complete picture, including the regulatory cues that may act in this process. Also, I missed a correlation between the presence of the genes in the genome and the actual phenotype of the isolates. For example, the isolates have genes to produce rhamnolipids and biosurfactants, but do they actually produce them? That would not be so difficult to test.

Specific comments:

Line 78-79 - the more accurate experiment would have been RNAseq analysis along with the genome sequence in order to gain insights into function.

Line 73-75 - a reference should be added to the text.

Line 90 (Materials and Methods) a more detailed description of the isolates would be suitable.

**Reviewer#4:**

1.In the section of Comparative genomics and Gene gain and loss analysis, it is necessary to describe more clearly the specific methods and steps of genome comparison and gene gain and loss analysis, and how to ensure the reliability and accuracy of the results.

2.The Results and discussion section requires more detailed description of the findings, especially in terms of functional analysis and genome comparison, which requires more explanation and discussion.

3.The potential applications of these bacteria in crude oil degradation and environmental remediation need to further explored in the discussion section, as well as possible future research directions.

4.Experimental validation was performed to further validate the results of the genomic analysis and to deepen the understanding of the role of these bacteria in real environments.

---

## [Author Response · Author response to Decision Letter 1]

17 Apr 2024

Reviewer #1

In this manuscript, a study on a bacterial community isolated from a hydrocarbon contaminated marine site was conducted. The genome of four isolates was investigated in detail and a plethora of information was obtained, focusing on identification of genes encoding for proteins involved in hydocarbons degradation, allowing to shed light on the role isolated strains may have in oil hydrocarbons degradation and giving a base for possible bioremediation uses.

This study offers an interesting view of the role different members of a microbial consortium may perform, thus conducting to an efficient degradation of an organic complex mixture. Oil hydrocarbons with their aliphatic and aromatic components, their hydrophobic nature, their scarce bioavailability, can be used as carbon and energy source by heterotrophic bacteria.

The manuscript is well written and can give insights for future research.

Line 25: ‘… 138 bacterial species …’ change to ‘… 138 bacterial stains …’

Thank you for your observation. L25, now L26, has been modified by changing "species" to "strains."

Biosurfactants allow better bioavailability of hydrophibic substrata as hydrocarbons, in the meantime, adhesion of bacterial cells to hydrocarbons can represent an important aspect favoring bacterial degradation of hydrophibic substrata. Investigation in this context could add information on the efficiency of these isolates in degrading hydrocarbons in bioremediation processes.

Reviewer #2

1. The manuscript is well written; however, the author needs to address the following comments

2. Improve the keywords, that should be reflected the nature of the work

Thank you for the comment. The new keywords in L38-39 are “marine bacterial consortium, hydrocarbon-degradation genes, functional diversity, comparative genomics, pangenome analysis”.

3. Introduction part needs enhancement.

The introduction was improved emphasizing the environmental impact of oil spills (L41-43). It also tightens the connection between HC pollution and bioremediation, focusing on the role of bacteria as a natural solution (L44-48). Additionally, complex sentences have been rephrased and redundant wording removed, improving overall readability (L60-80).

4. What are the potential sources of contamination, and how was this contamination assessed?

Regarding the first question, the new Introduction highlights in L41: "Crude oil spills, a consequence of anthropogenic activities in the ocean" acknowledging the human role in these environmental disasters. 

For the second question, oil spill contamination can be assessed through different methods like visual observation, chemical analysis of water and sediment samples, and remote sensing (NOAA 2020; Mauseth and Parker 2011). Our present research focuses on the bioremediation stage that follows the initial assessment. By understanding the genetic content of HC-degrading bacteria, we can develop more effective strategies to utilize these natural clean-up crews for future spills.

References

Mauseth, Gary S., and Heather Parker. 2011. “Natural Resource Damage Assessment.” In Oil Spill Science and Technology, 1067–82. Elsevier.

NOAA. 2020. “Oil Spills.” August 1, 2020. https://www.noaa.gov/education/resource-collections/ocean-coasts/oil-spills.

5. Check the typological error and language quality.

Thank you for your observation, the text was revised again, and we believe the errors were corrected.

6. What was the specific methodology used to isolate the bacterial strains or give any references for isolation.

Thank you for your question regarding the methodology used to isolate the bacterial strains. We apologize for not including this information in the previous draft. We have added a section under "Consortium strains and nucleic acid extraction" detailing the isolation process (L87-91). Briefly, the strains were isolated from the Gulf of Mexico using enrichment culture techniques followed by purification through streaking [reference the isolation method in reference 15]. We hope this clarifies our methodology.

7. Rewrite the Consortium strains and nucleic acid extraction

 Done as we described in the previous question.

8. How to conform the biosurfactant production of this bacterial strain (any specific analysis).

While our study focused on predicting the metabolic potential for hydrocarbon degradation and biosurfactant production, confirming biosurfactant production through specific analyses (e.g., droplet collapse assay, and surface tension measurement) would be a valuable addition to future research but was not part of the main scope of this work. We added this suggestion in L575-578.

9. The images are not clear give high-DPI quality images.

Done. The new version has all images in high quality according to the PACE tool by PLoS.

10. Check the References are placed correctly.

The text has been reviewed, and we believe the References are placed correctly.

Reviewer #3

The manuscript entitled “A comparative genomic study of a hydrocarbon-degrading marine bacterial consortium” by Pardo-López et al. presents a comparative analysis of four bacterial genomes. The marine bacteria were isolated in the Gold of Mexico in a previous paper published by the group (Rojas-Vargas J, Adaya L, Silva-Jiménez H, Licea-Navarro AF, Sanchez-Flores A, Gracia A, et al. Oil-degrading bacterial consortium from Gulf of Mexico designed by a factorial method, reveals stable population dynamics. Frontiers in Marine Science. 2022;9. doi:10.3389/fmars.2022.962071). Bacteria were used in consortium to degrade hydrocarbons in the previous report with success. In this paper the group presents the sequence of the bacterial genomes and analysis. The genome sequence of the 4 isolates allowed 3 important conclusions: (i) a taxonomic study and a phylogenetic reconstruction; (ii) the genome sequences allowed the identification of the genes related to alkane, aromatic and biosurfactant biosynthetic pathways in each isolate; and (iii) the gain and loss of the genes involved in hydrocarbon metabolism. From the knowledge and analysis of the sequences the authors propose complementary metabolic functions for the bacteria which would explain the higher hydrocarbon assimilation in the consortium compared to the individual bacterium. 

I find it important to gain knowledge of the genomes in a technical way, however, it would be much more interesting if the authors would have done a global transcriptional profile of the bacterial consortium, since the mere presence of the genes in the genomes do not guarantee its expression and function. The authors would have a much more precise and complete picture, including the regulatory cues that may act in this process. 

We agree with the Reviewer that a global transcriptional profile would have provided more detailed insights into which genes are being expressed and their function in the consortium in the context of hydrocarbon degradation.

However, an analysis of gene expression levels is not part of the objectives of the work, which focuses on comparative genomics and predicting the metabolic potential for hydrocarbon degradation and biosurfactant production based on in silico analysis of gene content (L23-26, L77-80). We have added in L572-575 the analysis of transcripts as future research directions. In particular, a transcriptomic analysis implies some technical challenges due to ribosomal depletion which to our knowledge probes to do so are not 100% compatible with the species that we are working with in this project, and it will require time to adjust that part to get the best results.

We also changed some expressions to strengthen the aim of the article. e.g. like “Alkane degradation metabolism” were replaced by “Alkane degradation potential” in the subtitles of section “Presence and refined annotation of HC-degrading genes” (L377, L412, L475); or “The functional analysis revealed potential metabolic differences…” (L588), now reads “The functional annotation analysis revealed potential metabolic differences…” 

Also, I missed a correlation between the presence of the genes in the genome and the actual phenotype of the isolates. For example, the isolates have genes to produce rhamnolipids and biosurfactants, but do they actually produce them? That would not be so difficult to test.

We appreciate the reviewer's comment. While confirming biosurfactant production through specific assays (e.g., droplet collapse assay and surface tension measurement) would be a valuable addition to future research, it was not within the scope of the present study. We have incorporated this suggestion in L575-578, highlighting the importance of such analyses for a more complete understanding of the consortium's capabilities.

Specific comments:

Line 78-79 - the more accurate experiment would have been RNAseq analysis along with the genome sequence in order to gain insights into function.

Thank you for your comment. We agree that RNAseq analysis would provide a more precise understanding of gene function compared to solely relying on gene content. To address this point and avoid any confusion, we have revised the introduction and the paragraph containing the lines you referenced (now L75-83). The new introduction now clarifies that our initial aim was to identify genes encoding proteins crucial for hydrocarbon metabolism, and to compare the genomes of our four bacteria with others within the same genera to understand how they evolved this genomic content.

Line 73-75 - a reference should be added to the text.

The reference was added (L75-76).

Line 90 (Materials and Methods) a more detailed description of the isolates would be suitable.

We have added a section under "Consortium strains and nucleic acid extraction" detailing the isolation process (L87-91).

Reviewer #4

1. In the sections of Comparative genomics and Gene gain and loss analysis, it is necessary to describe more clearly the specific methods and steps of genome comparison and gene gain and loss analysis, and how to ensure the reliability and accuracy of the results.

Thank you for your comment. We have modified the writing in both sections. Briefly, in the “Comparative genomics”, we simplified the text describing the selection criteria for the reference genomes instead of the more complex criteria in version 1 (L161-162) and enhanced readability (L163-166). In the “Gene gain and loss analysis”, we clarified the inputs used to run the COUNT software and made the paragraph more informative (L171-177).

The accuracy of our results relies on two factors: the quality of the data we used and the settings we chose for each software tool. In the comparison of genomic blocks, we opted to use complete genomes of closely related organisms (L161-162). We employed Sibelia with the default parameters (L167) recommended by the authors for this purpose (-s loose -m 5000) (Minkin et al. 2013).

We also believe that our approach for the gene gain and loss analysis was well-suited. We used a phylogenetic tree (based on 92 core bacterial genes) predicted by UBCG, a well-established method for inferring relationships between taxa (link to citations). Additionally, we employed Proteinortho (same parameters as HC-degradation gene annotation) to identify the presence/absence of orthologous genes (L174-176). This ensured consistency with the HC-degradation gene analysis in COUNT. Within COUNT, we applied the Wagner parsimony model (L176-177), which efficiently analyzes datasets of our size and considers all possible gain/loss events throughout evolutionary history (Csűrös 2008). Finally, we used a Poisson distribution at the root of the tree (L177) to account for potential variation in gene gain events at the origin of lineages.

References:

Csűrös, Miklós. 2008. “Ancestral Reconstruction by Asymmetric Wagner Parsimony over Continuous Characters and Squared Parsimony over Distributions.” In Comparative Genomics, 72–86. Springer Berlin Heidelberg.

Minkin, Ilya, Anand Patel, Mikhail Kolmogorov, Nikolay Vyahhi, and Son Pham. 2013. “Sibelia: A Scalable and Comprehensive Synteny Block Generation Tool for Closely Related Microbial Genomes.” In Lecture Notes in Computer Science, 215–29. Lecture Notes in Computer Science. Berlin, Heidelberg: Springer Berlin Heidelberg.

2. The Results and discussion section requires more detailed description of the findings, especially in terms of functional analysis and genome comparison, which requires more explanation and discussion.

In the sections of Comparative genomics and Gene gain and loss analysis, it is necessary to describe more clearly the specific methods and steps of genome comparison and gene gain and loss analysis, and how to ensure the reliability and accuracy of the results.

Thank you for your comment. We have modified the writing in both sections. Briefly, in the “Comparative genomics”, we simplified the text describing the selection criteria for the reference genomes instead of the more complex criteria in version 1 (L161-162) and enhanced readability (L163-166). In the “Gene gain and loss analysis”, we clarified the inputs used to run the COUNT software and made the paragraph more informative (L171-177).

The accuracy of our results relies on two facts: the quality of the data we used and the settings we chose for each software tool. In the comparison of genomic blocks, we opted to use complete genomes of closely related organisms (L161-162). We employed Sibelia with the default parameters (L167) recommended by the authors for this purpose (-s loose -m 5000) (Minkin et al. 2013).

3. The potential applications of these bacteria in crude oil degradation and environmental remediation need to be further explored in the discussion section, as well as possible future research directions.

Thank you for the reviewer's comments on potential applications and future research directions. In response, we added a subsection “Further directions” (L562-578) in the “Results and discussion” section. There we incorporated three suggestions:

- Potential application as a consortium in bioremediation strategies like bioaugmentation (L563-564)

- Scaling-up experiments using mesocosms, inoculation dosage strategies, or the study of using a lyophilized consortium (L566-570)

- A potential transcriptomic analysis to provide more detailed insights into gene expression and function within the consortium during hydrocarbon degradation (L572-575).

- Specific assays to confirm biosurfactant production by this bacterial strain (L575-578).

4.Experimental validation was performed to further validate the results of the genomic analysis and to deepen the understanding of the role of these bacteria in real environments.

Yes, experimental validation was confirmed through a previously published study (reference 15 in our manuscript).

---

## [Editor Report · Decision Letter 2]

24 Apr 2024

A comparative genomic study of a hydrocarbon-degrading marine bacterial consortium

PONE-D-23-38809R2

Dear Dr. Pardo-López,

We’re pleased to inform you that your manuscript has been judged scientifically suitable for publication and will be formally accepted for publication once it meets all outstanding technical requirements.

Kind regards,

Pankaj Kumar Arora, Ph.D

Academic Editor

PLOS ONE
---

## [Editor Report · Acceptance letter]

30 Jul 2024

PONE-D-23-38809R2 

PLOS ONE

Dear Dr. Pardo-López, 

I'm pleased to inform you that your manuscript has been deemed suitable for publication in PLOS ONE. Congratulations! Your manuscript is now being handed over to our production team.

Kind regards, 

on behalf of

Dr. Pankaj Kumar Arora 

Academic Editor

PLOS ONE